

# 3D texture-based face recognition system using fine-tuned deep residual networks

Siming Zheng[1], Rahmita Wirza OK Rahmat[2], Fatimah Khalid[3] and Nurul Amelina Nasharuddin[4]

[1] CASD, Department of Multimedia, Putra Malaysia University, Sedang, Malaysia
[2] C1-103 CASD, Department of Multimedia, Putra Malaysia University, Sedang, Malaysia
[3] C2-32, Department of Multimedia, Putra Malaysia University, Sedang, Malaysia
[4] C2-45, Department of Multimedia, Putra Malaysia University, Sedang, Malaysia

## ABSTRACT

As the technology for 3D photography has developed rapidly in recent years, an enormous amount of 3D images has been produced, one of the directions of research for which is face recognition. Improving the accuracy of a number of data is crucial in 3D face recognition problems. Traditional machine learning methods can be used to recognize 3D faces, but the face recognition rate has declined rapidly with the increasing number of 3D images. As a result, classifying large amounts of 3D image data is time-consuming, expensive, and inefficient. The deep learning methods have become the focus of attention in the 3D face recognition research. In our experiment, the end-to-end face recognition system based on 3D face texture is proposed, combining the geometric invariants, histogram of oriented gradients and the fine-tuned residual neural networks. The research shows that when the performance is evaluated by the FRGC-v2 dataset, as the fine-tuned ResNet deep neural network layers are increased, the best Top-1 accuracy is up to 98.26% and the Top-2 accuracy is 99.40%. The framework proposed costs less iterations than traditional methods. The analysis suggests that a large number of 3D face data by the proposed recognition framework could significantly improve recognition decisions in realistic 3D face scenarios.

Corresponding author
Siming Zheng,
gs53626@student.upm.edu.my

## INTRODUCTION

With the rapid development of the Internet, smart computing equipment and social networking applications are increasingly used. There are hundreds of millions of 3D images uploaded every day to platforms such as Snapchat and Alipay, on which a large number of 3D face images are generated. Three main problems in creating 3D face recognition systems that many researchers report are the 3D face pose, illumination changes, and variations in facial expression. Extracting better features are a key process for 3D face recognition (*Bagchi, Bhattacharjee & Nasipuri, 2015*; *Zhang et al., 2016*; *Nagi et al., 2013*; *Wang et al., 2015*; *Zhu et al., 2017*). Furthermore, shallow learning (such as machine learning) including only one or no layer of hidden units leads to lack of ability to deal with large-scale data. These challenges have caused persistent problems for the robustness and reliability of such

systems, which has driven many researchers to use deep learning for 3D face recognition tasks.

When deep learning methods are applied in realistic 3D face scenarios, two challenges confronted are as follows: firstly, the accuracy becomes unstable as 3D face images are added; this is because different deep learning networks have different generalization ability extracting images features. When processing a large number of image data, the deeper the layers of deep learning model are, the more problems such as gradient vanishing and gradient exploration will be caused. Secondly, as more and more complex deep learning models will be applied to the actual scenario, the recognition rate may be affected by the depth of a complex model. In this article we explore both issues. How to recognize a large number of 3D face graphics with high precision is the main task of this article.

In this work, the primary objective of the approaches we propose is to create a 3D textures-based end-to-end face recognition system with a high recognition accuracy, a satisfied performance and robustness while remaining practical. In this system, we have developed a residual neural network model base on ResNet for the 3D face recognition task. This model is fine-tuned with different depths using HOG featured 3D face textures. The primary aim is to solve problems of gradient vanishing and gradient exploration. We trained fine-tuned ResNet models with different depths using HOG based 3D texture images to maintain faster calculations and a high accuracy of image growth.

The remainder of this work is prepared as follows. 'Related Works' reminds the related work. 'Materials & Methods' presents methodology of extraction of HOG features and the fine-tuning ResNet model. 'Experiment' shows the experimentation, results and discussion is described in 'Results and Discussion'. The conclusions are finally stated in 'Conclusions'.

## RELATED WORKS

Deep learning algorithms have received increasing attention in the face recognition field, and many researchers discovered the importance of studying 3D face recognition (*Maiti, Sangwan & Raheja, 2014*; *Min et al., 2012*; *Pabiasz, Starczewski & Marvuglia, 2015*; *Porro-Munoz et al., 2014*; *Hu et al., 2017*; *Sun et al., 2015*; *Wu, Hou & Zhang, 2017*; *Tang et al., 2013*; *Zhang, Zhang & Liu, 2019*). On one hand, extracting 3D face information is the key step in 3D face recognition: effective face detection and alignment can increase the overall performance of 3D face recognition, which is critical in both security and commercial 3D face recognition systems. On the other hand, researchers have proposed some methods for exploiting and exerting the deep learning for 3D face recognition, and they have demonstrated that the performance of deep learning systems is significantly better than that of machine learning method in the case of a large amount of 3D images.

In recent years, the convolutional neural network (CNN) models have been used for 3D face recognition. *Hu et al. (2017)* has proposed a method of customizing convolutional neural networks. Her CNN's layer configuration uses the same principle to design based on the LeCun model (*LeCun et al., 1989*). The structure of her model, called CNN-2, comprises one convolutional layer, one pooling layer, and a 5×5 filter. However, this structure cannot effectively extract and analyze 3D face data. When the learning rate rose

from 0.034 to 0.038, the classification accuracy increased from 84.04% to 85.15%; while, the accuracy dropped to 81.31% when the learning rate rose to 0.042. Furthermore, using a $7 \times 7$ filter increased the classification accuracy significantly to 84.75% with a learning rate of 0.034.

In a follow-up study, *Sharma & Shaik (2016)* have proposed a new methodology for face recognition with a higher accuracy of approximately 98%. They suggested a customized CNN model, including an input layer, a convolutional layer, a pooling layer, and a fully connected layer. They use the above method to recognize the 3D image with resolution of 96×96. According to results, their face recognition system takes twenty epochs for converging the learning rate, which includes the training rate and the testing rate. Especially, the training losses can be decreased to about 0 before the 6th epoch.

Different methods have been proposed to recognize 3D face images. *Kim et al. (2017)* has developed the VGGNet neural network for dealing with 3D face data. The most representative features of face are extracted from the fine-tuned VGGNet model. The model includes two convolution layers and two fully connected layers with random initial weights, using the last fully connected layer with a softmax layer to accommodate the different sizes of the input images. The fine-tuned VGGNet model achieved an accuracy of 95.15% in the experiment.

*Nagi et al. (2013)* has developed the face alignment algorithm based on the methods of geometric invariants, local binary pattern (LBP), and k-nearest neighbor (kNN). The face landmarks model (22 key points) is used to detect the human face, and the LBP method is used to crop the 3D face areas. The method of kNN calculates the distance between each input data and the training sample, obtaining the k images closest to the training sample. Finally, proposed statistical methods are used to classify and recognize the images. The results show that the model can reach 91.2% in the recognition rate; however, it declined to 84% as the number of datasets increases.

*Soltanpour & Jonathan Wu (2017)* uses normal vector to study 3D face recognition. She proposed that more detailed distinct information can be extracted from the 3D facial image by using high-order LNDP method. By estimating the three components of normal vectors in x, y and z channels, three normal component images are extracted. The score-level fusion of three high-order LNDP3x, LNDP3y and LNDP3z are used to improve the recognition performance. Experiments use SIFT-based strategy for matching the face features. The results of this study indicate that fusion LNDP3xyz outperforms descriptors, effectively improving the 3D recognition rate to 98.1%.

The study by *Kamencay et al. (2017)* offers probably the most comprehensive empirical analysis of 3D face recognition. In an attempt to build practical and robust face recognition systems, he proposed three main types of layers for CNN architectures: the convolution layer, the pooling layer, and the fully connected layer. He also proposed three machine learning methods for face recognition, such as Principal Component Analysis (75.2%), Local Binary Patterns Histograms (78.1%), and kNN (71.5%). The proposed customized CNN for 3D face recognition outperforms the above machine learning methods, which reaches the average accuracy of approximately 96.35%. The highest accuracy is 98.3% when 80% of the data was used for training model.

Recent advances in HOG feature extraction methods have facilitated investigation of face recognition. In *Singh & Chhabra*'s (*2018*) article, she suggests that HOG features can be used in the recognition system for improving the efficiency and the processing speed. The function of HOG features can capture the edge features that are invariant to the rotation and light. Owing to the fact that both texture and edge information is important for face representation. HOG features and SVM classifier-based face recognition algorithm is presented in *Santosh Dadi & Mohan Pillutla*'s (*2016*) research. His proposed model extracts the HOG features of the face from the image, these features can be given to any classifier. In the testing stage, the test image is obtained and fed to the SVM classifier, which is a non-probabilistic binary classifier and looks for optimal hyperplane as a decision function, for classification. The results show that this method has better classification accuracy for the test data set, about 92%. In addition, compared to the method using standard eigen feature and PCA algorithm as a baseline, SVM also possesses an improved face recognition rate of 3.74%.

To investigate the effect of utilizing HOG features in the CNN model, (*Ahamed, Alam & Manirul Islam, 2018*) developed CNN learning models that using the HOG features as input data to the training model. His model contains of several layers and each layer is repeatedly used, finally a deep neural network is constructed. In order to evaluate the proposed model, a set of images with 160 images are generated for testing the model performance. However, it leads to a low generalization ability since the data set trained by the model is small. The result shows that the accuracy is approximately 89% by using the constructed model.

In the experiment, we used the latest residual deep neural network (ResNet) and the fine-tuning method (*He et al., 2015*). Our preprocessing method uses HOG features of 3D face texture, different layers of ResNet are created during the experiment and whether decision making in face recognition process can be improved or not is investigated. We evaluated these approaches in the context of the same 3D face-recognition experiment as in (*Kamencay et al., 2017*), a more challenging task than the face identification task used in (*Ahamed, Alam & Manirul Islam, 2018*).

## MATERIALS & METHODS

The diversity of face poses raises difficulties for 3D face recognition. By detecting key points on the face, such as the tip of the nose, the corners of the mouth, and the corners of the eyes, the face image in an arbitrary pose can be converted into a frontal face image by affine transformation, after which the face features can be extracted, and an identification is performed. This approach shows that after alignment the features can be extracted with greater success, and the recognition accuracy is thus greatly improved. A schematic diagram of face detection and face alignment is shown in Fig. 1. There are three steps for preprocessing of 3D face recognition: 1. 3D face detection, 2. 3D face alignment. 3. 3D human face feature extraction. The first two phases are implemented by using the open-source tool provided by the Dlib, which can monitor the key points of the face real time to obtain the position and posture of the face. Then we developed a module for extracting the HOG features based on 3D face texture images. Key points of the face

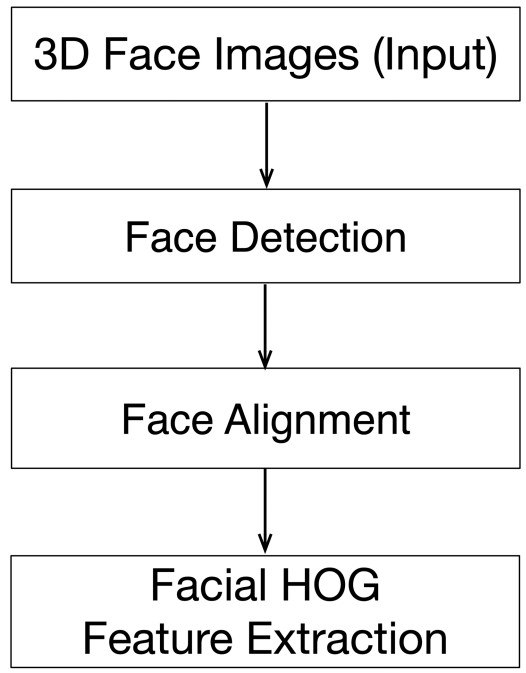

**Figure 1** **The pre-processing of 3D face textures.**

are detected using the conditional local neural fields algorithm (*Baltrusaitis, Robinson & Morency, 2013*; *Simonyan & Zisserman, 2015*).

## Facial detection and landmarks selection

All 3D images need to be processed before the processing of recognition in order to reduce image noise and redundancy, as shown in Fig. 2. The first step of 3D face recognition is face detection and alignment. We use pre-trained facial landmark detector from the Dlib library, which is used to estimate and predict the location of sixty-eight key points on the human face.

Based on the geometric invariant method, these facial points are marked on the 3D facial images (*Baltrusaitis, Robinson & Morency, 2016*; *Jourabloo & Liu 2015*; *Song et al., 2017*), the subgraphs of A and C is the original 3D images, and the 68 key point distributions are indicated as B and D in Fig. 2 on the right side. These points, including the dominant facial features, such as the tip of the nose, the corners of the mouth, and the corners of the eyes, which are used for further feature extraction and geometric calculations in the recognition stage.

## The features of histogram of oriented gradient

In the feature extraction process, we usually try to find the invariant properties and characteristics so that the extraction results do not change significantly due to the specified conditions, this means that the goal of recognition is to find useful discriminative information not the position and size. Regardless of the different changes in the shape and

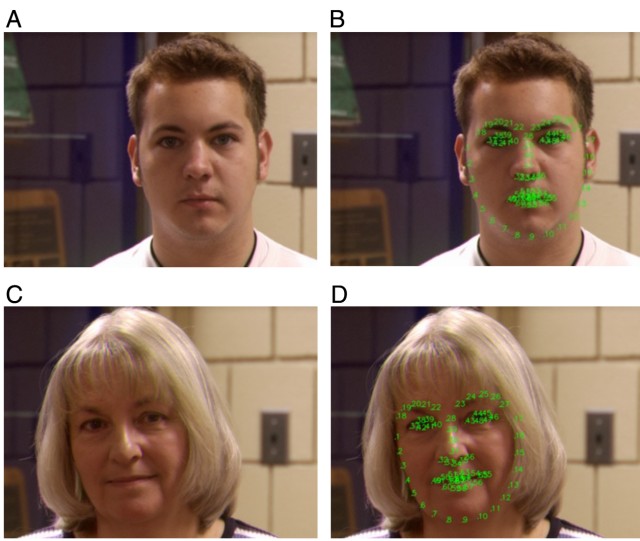

**Figure 2** **Facial landmarks (68 key points) of Face Recognition Grand Challenge Version 2(FRGC v2.0).** The subgraphs of A and C show that the camera takes images of the subjects from different angles. Note that face coordinates are located by using 68 green key points in the subgraphs of B and D.

appearance of the image, we should find reliable and robust discriminative information for improve the recognition rate.

In the field of image processing and computer vision, texture analysis and extraction have a rich history. A method called the Histogram of Oriented Gradient (HOG) has received extensive attention. The core idea of the HOG method is to describe the texture of the detected object by the gradient or distribution of edge directions. Its function is to capture the edge or gradient structure from the image, which is characteristic for the representation of local texture. The benefit of this feature is relatively less affected by the appearance and shape. Essentially, it forms a template and uses learning models to effectively promote recognition.

The HOG descriptor can extract important features from 3D images (*Santosh Dadi & Mohan Pillutla, 2016*; *Kumar, Happy & Routray, 2016*). It captures the local texture information well and has good invariance to geometric and optical changes. Firstly, the target image is divided into small connected regions, which call the cell units. Then, the gradient or edge direction of each pixel in the cell unit are acquired. Finally, the histograms can be combined to form a feature descriptor. In this section, the HOG feature is used as a means of feature extraction in the process of recognition, the purpose is to combine the discriminative 3D face feature in the recognition phase, the specific implementation steps are as follows.

(1) Color and gamma normalization

To reduce the influence of lighting factors, the entire image needs to be normalized in the first step. A compressing process that can effectively reduce shadows, colors and illumination variations of the image, because this information did greatly increase code complexity and demanded the higher performance of processor. At the same time, the

gray image is normalized by Gamma formula. By smoothing part of noises, the influence of local strong light on gradient calculation is reduced. The $\gamma$ is the symbol of Gamma and its value is 1. The formula of gamma compression Eq. (1) is shown below.

$$I(x,y) = I(x,y)^\gamma \tag{1}$$

(2) Gradient computation

The gradient value of each pixel position is calculated in this step. The derivation operation can capture contours, human shadows, and some texture information, which further weakens the influence of illumination. In the operation of computing image gradient, the gradient direction is key to HOG algorithm. The function of $H$ is used for calculating of Histogram of Oriented Gradient. Each pixel point of the transverse gradient $G_x(x,y)$ and the longitudinal gradient of the $G_y(x,y)$ is calculated. It defined as Eqs. (2) and (3).

$$G_x(x,y) = H(x+1,y) - H(x-1,y) \tag{2}$$
$$G_y(x,y) = H(x,y+1) - H(x,y-1) \tag{3}$$

(3) Creating the orientation histograms

The algorithm needs to finish some operations that calculating the direction gradient of the smallest interval. At the beginning, the 3D image is divided into several intervals with different sizes. Starting from the smallest interval, the gradient direction of all the pixels are contained in each interval, which are weighted by the magnitude. The gradient direction with the largest value represents the gradient direction in the current interval. Finally, the gradient magnitude $G(x,y)$ of each pixel point is calculated according to Eq. (4).

$$G(x,y) = [G_x(x,y)^2 + G_y(x,y)^2]^{1/2} \tag{4}$$

Furthermore, the specific operation in the equation above is that the $G(x,y)$ ranges from $-90°$ to $90°$. Vectors can be evenly divided into nine intervals because each interval is $20°$. This means that nine intervals consist of a total of nine feature vectors in a cell. Four cells can form a block, so each block includes 36 feature vectors. In this way, the feature vectors of all cells in a block are concatenated to form the HOG features.

(4) Computing the directional gradient histogram

In the final step of the merging process, the algorithm uses weighted voting to combine the order of all blocks from smallest to largest. In some cases, the algorithm eliminates some detailed features, which are represented by the small amplitude gradient in the intervals. The rest of the blocks are merged into a maximum gradient pattern, in which contains the important representative features in the 3D images. Gradient direction a(x, y) of the pixel point are calculated according to Eq. (5).

$$\alpha(x,y) = tan^{(-1)}[G_y(x,y)/G_x(x,y)] \tag{5}$$

(5) Creating the orientation histograms

After completing above works, the algorithm generated an orientation histogram for the input 3D face image. In this experiment, we make the pixel of $16 \times 16$ constitute a

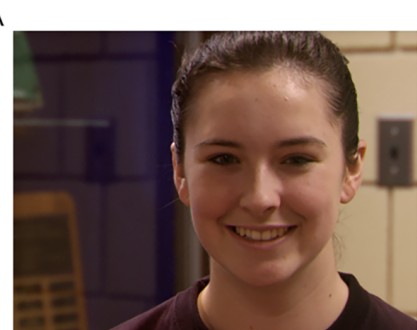

Original 3D face image

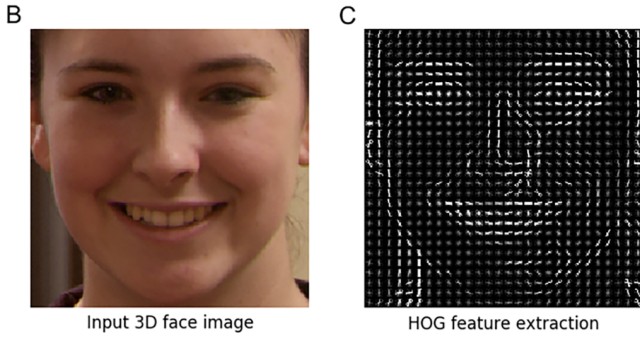

Input 3D face image      HOG feature extraction

**Figure 3** **The processing of HOG feature extraction.** (A) Original 3D face image (B) input 3D face image; (C) HOG feature extraction.

block in an image with $224 \times 224$ image, the stride of scanning window is $8 \times 8$, then there are 27 scanning windows in the horizontal and vertical directions in a 3D texture image. Therefore, each 3D texture image ($36 \times 27 \times 27$) has 26,244 dimensional vectors that can form a complete edge orientation histogram.

After the preprocessing, the face image with extracted HOG features of 3D textures is input into our fine-tuned ResNet classification model. The information contained in the original image is compressed and adjusted, which greatly improves the performance of the subsequent feature extraction network in ResNet neural network. Finally, the whole processing of HOG feature extraction for 3D face image is shown in Figs. 3A, 3B and 3C.

We also demonstrate generation of HOG feature vectors for specific person with different expression, scenarios and various illumination changes. The images based on HOG features extraction are shown from the Figs. 4F to 4J, which are separately correspond to the reprocessing aligned images from Figs. 4A–4E.

## The architecture of ResNet neural networks

Convolutional neural networks with multiple layers have several advantages in the research of image classification. The deep network uses a form of end-to-end neural network that automatically integrates the low, medium, and high-level features, and then transmits all of these features to the classifier, extracting different depth features by stacking layers with different depths. Another benefit of the CNN network is that the convolutional layer can

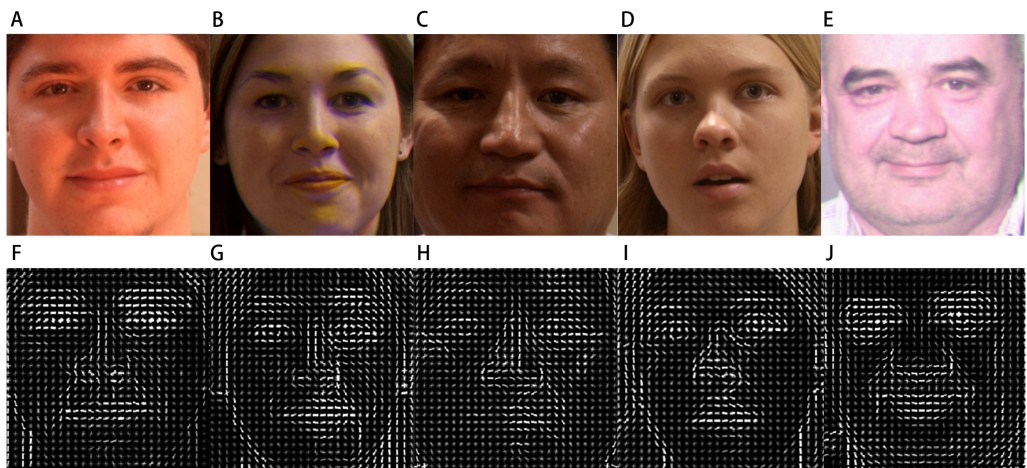

**Figure 4** **The HOG features of various 3D face images.** The subject images (A, B, C, D, and E) were taken under the different conditions of illuminations, expressions, sexes, and ages. Depending on the parameters in the HOG calculation process, the experimental results ensure that the acquired images (F, G, H, I, and J) reveal the discriminative information related to the texture of the original face images.

retain local spatial patterns which may be appropriate for image related tasks (*Zeiler & Fergus, 2014*). Recent research has shown that the depth of the CNN network is critical to model performance (*Szegedy et al., 2014*; *Cheng et al., 2017*). The results of their studies show that the deep convolutional neural network achieves superior performance and significant improvements.

In general, the deeper the neural network is, the worse the recognition performance will be. One major issue in early 3D face recognition research is caused by the use of an error back-propagation algorithm (*Lawrence & Lee Giles, 2000*), which includes the weight coefficient, the derivative of the activation function, and the activation value in the partial derivative. When the number of layers is large, these values are multiplied, easily leading to the vanishing gradient and exploding gradient problems (*Pascanu, Mikolov & Bengio, 2012*; *Hanin, 2018*). Therefore, it is difficult to ensure high-accuracy in the case of growth of 3D face data. In the paper by *He et al. (2015)*, he proposed a theory of deep residual learning, which adopted an approach of shortcut connection to avoid the issues mentioned above. The ResNet architecture for a 152-layer network (a) and a residual block (b) are shown in Fig. 5 above. A residual block with a connections layer can skip a specific layer in the network. The advantages of short connections are that it can reduce the problem of gradient disappearance, thus making the network converge faster and reducing parameters. ResNet-152 also uses the batch normalization operation between each convolution and activation. It allows the researcher to build increasingly deep networks, which have high recognition abilities.

## The fine-tuned ResNet neural network model

In deep neural networks, the function of the first layer of training on images is similar to the Gabor filters and color spots operations. First-layer features are not used for specific

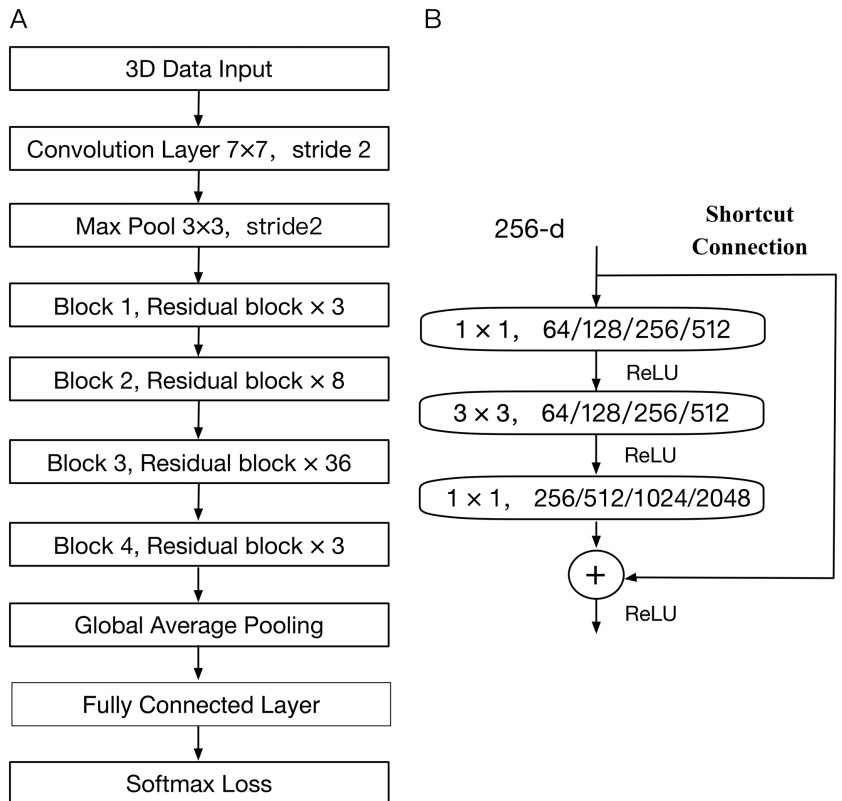

**Figure 5** **The architecture of ResNet model.**

data sets or specific tasks but for general ones, because they are applicable to common data sets and tasks. Image features are eventually transmitted from general to specific by the last layer of the network (*Yosinski et al., 2014*). In the big data scenario, we introduce the fine-tuning method in the ResNet neural network, which can greatly shorten the training time, efficiently improve the training loss, and have a stronger generalization ability for getting a good result.

(1) Fine-tuning method

The fine-tuning method can be used to flexibly adjust the architecture of the ResNet model in this 3D face recognition task (*Jung et al., 2015*). In our experiment, four pooling layers with the adaptive average pooling method have been reconstructed. By using the new architecture, it makes the input training data adaptive to the fine-tuned ResNet Model, and the computational complexity of the model is reduced. A softmax layer is created after the fully connected layer to implement the target data classification task of this experiment.

(2) Rectifier Linear Unit

The Rectifier Linear Unit (ReLU) is an important activation function in the ResNet structure, which can overcome the problem of gradient disappearance and speed up the time of training (*Fred & Agarap, 2018*). A ReLU function maps the input value $x$ to 0 if it is negative, and keeps its value unchanged if it is positive, the main ReLU calculation

| Structure | Output Size | Layers |
|---|---|---|
| Conv layer 1 | ( Input size: 224 x 224 ) 112x112 | ( Input channels: 1 ) 7x7, 64, stride 2 + BatchNorm |
| Conv layer 2 | 56x56 | 3x3 max pool, stride 2<br><br>1x1, 64<br>3x3, 64    x3 + BatchNorm<br>1x1, 256 |
| Conv layer 3 | 28x28 | 1x1, 128<br>3x3, 128    x4 + BatchNorm<br>1x1, 512 |
| Conv layer 4 | 7x7 | 1x1, 512<br>3x3, 512    x3 + BatchNorm<br>1x1, 2048 |
| | 1x1 | Adaptive Average Pooling |
| | 1x1 | Fully Connected Layers |
| | 1x1 | Softmax(466) |
| Model Output | | |

**Figure 6** **The fine-tuned ResNet neural network.**

expression (Eq. (6)) is shown below.

$$f_{relu}(x) = max(0, x) \qquad (6)$$

The convolutional neural network architecture is mainly followed with a combination of all the methods described above. The architecture starts from the input layer of training images and is followed by the convolution layer with the optimum weight and bias for the feature layer. In order to reduce the internal covariate shift (*Ioffe & Szegedy, 2015*) in the deep neural network, the batch normalization algorithm is also added to each convolutional layer to perform the operations of normal normalization and affine transformation on the input of each layer. Finally, our fine-tuned ResNet model were constructed with the proposed method, and the parameters of each convolutional layer are represented in Fig. 6.

In this fine-tuned ResNet model, the layer of adaptive average pooling emphasizes the down-sampling of the overall feature information, its purpose is to reduce the dimension of the feature and retain the effective information, it can integrate features in the feature maps from multiple convolutional layers and pooling layers so that the integrity of the information in this dimension can be more reflected. Through this process, both high-dimensional features and confidence scores can be obtained from each classification. The final full connection layer is used to synthesize the features extracted from the adaptive

average pooling, it can output the probability distribution by using the softmax regression method, which can be divided into more than 1,000 classifications for any tasks, and the value of this parameter was set to 466 in our experiment. In the above structure, Adam function is used as an alternative to the traditional Stochastic Gradient Descent (SGD) optimization algorithm which can iteratively update the weights based on the training data mainly to optimize the neural network and make the training faster.

## Datasets

This research received the approval from the University of Notre Dame (henceforth, UND), and the dataset of Face Recognition Grand Challenge version 2 (FRGC-v2) in January 2019. This experiment was performed on FRGC-v2 (*Flynn, 2006*), which is a large number standard face image dataset containing over 50,000 high-resolution 2D and 3D face images, which divided into training and validation partitions in a laboratory setting. Training partition is designed for training algorithms, and validation partitions is used to evaluate the performance of a method in a laboratory environment. All the images were captured by a Minolta Vivid 900/910 series scanner. The datasets used belong to Experiment 3 of FRGC-v2. The experimental 3D face dataset includes 4,007 images of 466 people with different lighting and facial expressions The aim of this dataset is to test the ability to run experiments on large datasets (*Phillips et al., 2005*).

## Graphics processing unit

The efficient parallel computing of the graphics processing unit (GPU) makes up for the slow training of deep neural networks (*Chen et al., 2014*). Combined with the CUDA parallel computing platform, it allows the use of larger training datasets and deeper complex neural networks to extract deeper image features (*Huang et al., 2015*; *Singh, Paul & Dr. Arun, 2017*). The model of GPU used in this experiment is NVIDIA GeForce GTX 1080Ti. Its multi-core architecture includes thousands of stream processors, which can perform vector operations in parallel and achieve several times greater throughput in the application. This significantly shortens the calculation time. GPU has therefore been widely used by scientists in deep network learning.

The Fig. 7 shows the processing of our proposed framework. The detailed steps of the proposed framework are as follows: first, the image is detected by the 68 key points of facial landmarks method, and the main multi-channel 3D face regions are extracted, this preprocessing module achieved precision rate of 99.85% and effectively reduces image noise and redundancy from original images, and all images are rescaled to the $224 \times 224 \times 3$. Then, the edges and textures of 3D face images are enhanced by using the HOG method with custom parameters, the HOG face feature images based on 3D textures are obtained, which learned higher discriminative features from 3D face images. Next, fine-tuned deep residual model is proposed by using the HOG textures as the input images. Finally, we generated custom ResNet neural network model. Its image input size is adjusted to the pixel of $224 \times 224 \times 3$, and the structure and quantity of the middle layer in the model are reconstructed, all the operations are performed by using fine-tuning method.

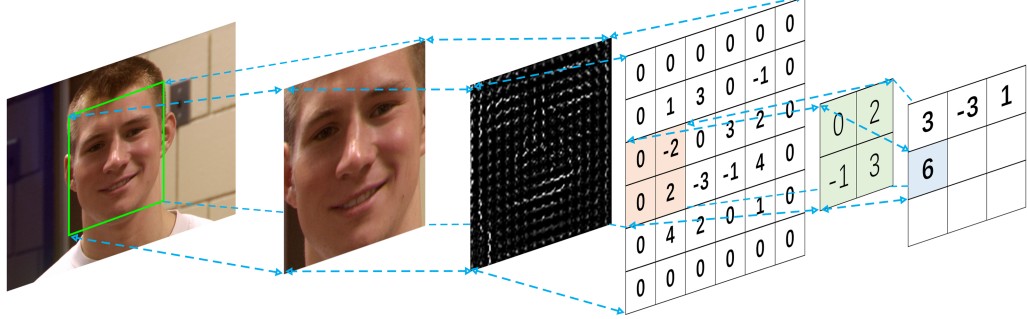

**Figure 7** **The fine-tuned ResNet feature extraction operation.**

## EXPERIMENT

Key point detection and alignment were carried out on the 3D raw face texture in succession. In the following steps, the dataset applied to the fine-tuned ResNet model comprises 3D face texture with HOG features. Finally, the proposed model conducted for 3D face recognition. In this experiment, an implementation of GPU accelerated training is adopted based on Python and the CUDA architecture, all the HOG-featured images were resized of $224 \times 224 \times 3$ pixel in the dataset, the test model of fine-tuned ResNet with different depth layers (e.g., 50 layers, 101 layers, 152 layers) were then evaluated.

(1) Firstly, a convolution layer in fine-tuned ResNet architecture multiplies the $2 \times 2$ filter with a highlighted area (also $2 \times 2$) of the input feature map, and all the values are summed up to generate one value in the output feature map, as shown in Fig. 7.

(2) After the 3D data are processed through the first convolution layer, the next layer is max-pooling. The filter window of the max-pooling is moved across the input feature with a step size defined by the stride (the value of stride is 2 in the case of ResNet-152).

The advantage is that it can reduce errors and preserve more texture information. In the max-pooling, the maximal value is selected from four values in the filter window. The size of the detection region is $f \times f$, with a stride of $s$, so the output features $h'$ and $w'$ are given through the Eq. (7) below.

$$h' = [\frac{h-f+s}{s}], w' = [\frac{w-f+s}{s}]. \tag{7}$$

(3) The residual block consists of two convolution layers each a $1 \times 1$ filter and one convolution with a $3 \times 3$ filter. The $1 \times 1$ layer mainly reduces and restores dimensions, leaving the $3 \times 3$ layer a bottleneck with smaller input/output dimensions. Two $1 \times 1$ convolutions effectively reduce the number of convolution parameters and the amount of calculation. The residual block is used for ResNet-50/101/152.

(4) The fine-tuned ResNet model uses a global average pool and then categorizes 3D face images at the end of the network through fully connected layers. The global average pooling layer provides faster calculations with more accurate classification and fewer parameters. It serves to sum up all the values from the filter window and then average them, which can reduce errors and retain 3D background information of the image.

(5) Finally, the fully connected layer reassembles the previous local 3D features into a complete graph through the weight matrix. The classification $y$ is defined as follows:

$$y = f(W^T x + b) \tag{8}$$

(6) However, having a good neural network model in specified dataset does not necessarily imply that the model is perfect or that it will be reproduced when tested on external data. In order to make sure it is robust, reproducible and unbiased for testing future new datasets under non-ideal conditions, accuracy metrics is adopted to evaluate the fine-tuned ResNet model's performance.

The indicator accuracy is a measurement of the correct proportion of image classifications. This study is accurate in its ability to differentiate the 3D face recognition cases correctly. To estimate the accuracy of tests, the proportion of true positives (TP) and true negatives (TN) in all evaluated cases are calculated in this experiment. Mathematically, this can be stated as following.

$$Accuracy = \frac{TP + TN}{TP + TN + FP + FN} \tag{9}$$

The sub formula of TP+TN+FP+FN is the total number of observations. Moreover, the respective tests of Top 1 and Top 2 accuracy are used to evaluate the performance of our proposed model. The following hypothesis will be tested: the increasing network depth improves the accuracy of 3D face recognition. This study contributes to this developing area of research by exploring how different depth of our fine-tuned ResNet networks affect the outcome of 3D recognition.

## RESULTS AND DISCUSSION

Tensor Board is a data visualization tool that can be used to visualize computational graph structure, provide statistical analysis, and plot the values captured as summaries during the execution of computational graphs. In this research, different types of pre-trained ResNet neural network with the same structure but different depths are proposed. As shown in Fig. 8, the three-subgraph shown below are the Tensor Board graph of linear regression corresponding to the 50 (A), 101 (B), and 152 (C) layers of structure of the ResNet neural networks. The 3D face recognition rates of the three different structural models were recorded objectively in real time use of Tensor Board.

There were 21 times of training and testing in all test cases. According to the results of the ResNet50 testing model, the maximal accuracy was 97.02% for the validation set in the 21th epoch, which corresponds to the similar accuracy of 97.22% in the 6th epoch of the ResNet101 model. This accuracy rate is close to 97.10% in the same epoch in the ResNet152 model. The highest accuracies were 98.05% for ResNet 101 and 98.26% for ResNet 152.

The most accurate indicator of Top 2 can be used to further evaluate the performance of the trained ResNet model. In the ResNet 152 model, the recognition rate fluctuates at first and then becomes regular with the increase of test sets. The accuracy rate was maintained at an average of 99.30% for ResNet 152 after the 12th epoch. The results show that the

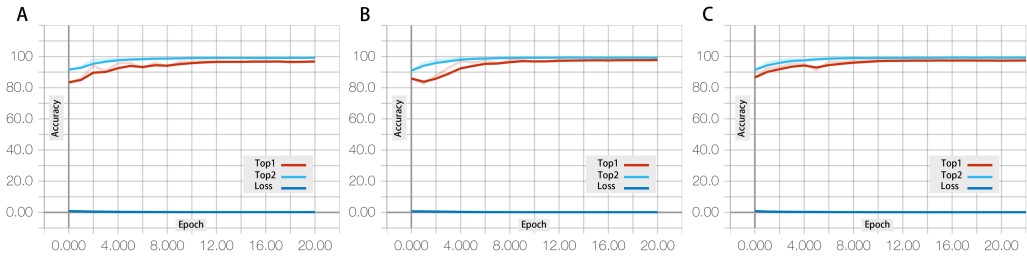

**Figure 8** **The accuracy rate of different layer numbers in the fine-tuning ResNet architecture.** The performances are shown for the fine-tuned ResNet-50 layer (A), the fine-tuned ResNet-101 model (B), and the fine-tuned ResNet-152 model (C).

ResNet model has strong generalization ability. The recognition rate reaches its peak of 99.40% in ResNet-152 in the 12th epoch of the 3D face recognition experiment.

This experiment explores the benefits and effect of different numbers of neural network layer through the fine-tuning method on 3D face texture recognition research with high accuracy. The Fig. 8 has shown that the increasing of layers of fine-tuned ResNet neural network model, the proposed framework can improve the accurate through the HOG method based on 3D face textures.

To eliminate the effects of interference factors, the 3D face dataset was processed beforehand (3D face detection, alignment, and HOG feature extraction). Studies have shown the importance of the fine-tuned convolutional neural network model (ResNet) with depth layers that have more highly discriminative features. The model is advantageous in that although the depth is significantly increased, the ResNet model is less complex with a higher accuracy rate. The Fig. 8 presents the inter-correlations among the three recognition rates of the ResNet model with different layers. The 3D face recognition rate is positively correlated with the number of layers in the ResNet model, which is also a principal factor determining the computing time.

With qualitative modes of enquiry employed, the experiments show that the proposed method achieves promising results, demonstrating that the ResNet-152 neural network model described in this paper can have a recognition accuracy of 98.26% (Top 1); compared with the most accurate, the accuracy of the second accurate test was improved by 1.14% (at 99.40%) with the FRGC-v2 datasets. Practical results proved the validity of the proposed method in 3D face recognition.

The classification performance of methods applied to the FRGC-v2 dataset seem superior to the seemingly impressive results of published studies utilizing different methods in Table 1 (*Hu et al., 2017*; *Sharma & Shaik, 2016*; *Soltanpour & Jonathan Wu, 2017*).

In previous researches, custom CNN is a commonly used deep learning algorithm used for 3D image recognition tasks. Firstly, in custom CNN network training, it is necessary to constantly adjust network parameters. The customized parameters such as weights and biases in CNN network results in a very slow convergence of training, and thus greatly increasing the training time and the number of epochs (*Hu et al., 2017*; *Sharma & Shaik, 2016*). In addition, when the dimension increases with the increase of data volume, it

**Table 1  Performance comparisons between the proposed method and state-of-the-art methods based on the FRGC-v2 dataset.**

| Method | Features | Classifier | Accuracy |
|---|---|---|---|
| Huiying Hu et al. | Raw image | Custom CNN-2 | 85.15% |
| S Sharma et al. | Constrained Local model | Custom CNN | 98% |
| Sima Soltanpour et al. | $LNDP^3_{xyz}$ Based Normal Component Images | SIFT-based matching Method | 98.10% |
| Proposed Methodology | HOG features | ResNet 152 layers ResNet 101 layers ResNet 52 layers and Fine-tuning | **Top1: 98.26%** **Top2: 99.40%** Top1: 97.77% Top2: 99.40% Top1: 97.02% Top2: 99.12% |

will lead to a curse of dimensionality problems and cause a drop in the performance of the classifier (*Soltanpour & Jonathan Wu, 2017*). The Fine-tuning method speeds up the convergence and shortens the training period, thus adapting to 3D processing. The purpose of multi-layer convolution is that the features acquired through one-layer convolution are often local, and the more layers there are, the more global the features will be acquired. Then, how to maintain good performance and improve accuracy is a key in larger numbers of 3D face recognition scene. Therefore, fine-tuning depth residual network is proposed based on HOG features to effectively solve the problem of large numbers of 3D face recognition.

To the best of our knowledge, our work is to examine a fine-tuned Deep Residual Networks model on the recognition task of FRGC-v2 dataset. To increase accuracy during ResNet training, several methods were considered in this paper: (1) a fine-tuning deep residual network was adopted, taking advantage of its intrinsic features, such as shortcut connection, weights sharing and pooling architectures, and these can be improved through the deepening of the network structure; (2) the number of layers is carefully designed with smaller filter size to avoid overfitting while there is sufficient capacity for the network to solve the complex large number classification problems (*Hawkins, 2004*; *Lawrence & Lee Giles, 2000*); (3) data extraction was performed via the HOG method at the image preprocessing that contains higher discriminative features in the 3D images. As a result, the proposed methods were well trained and yielded state-of-the-art classification accuracy.

## CONCLUSIONS

In this study, in-depth investigations were conducted on end-to-end 3D face textures recognition. We first review the previous studies on 3D face recognition and then summarize the critical research questions to be solved. The 3D face detection and alignment modules are implemented and flexibly applied in 3D face raw data, which achieves a precision rate of 99.85%. In addition, the detailed steps of the HOG extraction pattern were presented. 3D face images with HOG features can significantly minimize the descriptor size for reducing computation load and economizing the memory in the

recognition process. We trained the fine-tuned ResNet models combined with HOG features; the discriminative power of the deeply learned features can highly enhance recognition ability. This study implemented every important subcomponent, which can effectively reduce 3D image noise and greatly increase the robustness of our proposed recognition system.

The experiment showed that the degradation problem was efficiently solved by increasing the number of layers in our fine-tuned ResNet neural networks, which improves the recognition rate within a short time, and the accuracy is maintained at a certain level. However, although the performance of the algorithm is unexceptional in practical application, we think that several aspects of the model should still be studied and improved. Firstly, although the HOG algorithm is advantageous for less calculation time and faster detection speed, when the pose of the 3D face is changed drastically, there is a target loss in the face image, which leads to low processing efficiency. The 3D face alignment can be pre-processed with the CNN detection method, and a multi-processing or multithreading method can be used to speed up face alignment, which ensures that the pre-processing module can process data quickly. Secondly, the recognition rate may be adversely affected by certain conditions. For instance, the ResNet-152 model exhibited the phenomenon of overfitting, in which the accuracy rate dropped and remained at around 97% after the 9th epoch. This phenomenon is caused by two conditions: too few datasets and the excessive complexity of the neural network model. This can be solved by increasing the amount of 3D face data in the future works via a data augmentation method (*Perez & Wang, 2017*; *Wong, Gatt & Stamatescu, 2016*). This also shows that the ResNet network has a more powerful data processing capability for a large number of data. Overall, the development of large number 3D face recognition classification system is a challenging work, and there is still a long way to go to apply these theories and methods in large-scale scenes. The results suggest that fine-tuned deep residual networks classification approach based on HOG features will be a promising direction to improve 3D face recognition rate.

## ACKNOWLEDGEMENTS

I am indebted to Prof. Rahmita Wirza O.K. Rahmat for assistance with data collection, providing helpful discussions of the 3D face analyses. I thank Fatimah Khalid for her help in the depth analysis and comments on the face recognition theory, as well as Prof. Dr. Nurul Amelina Nasharuddin for fruitful advices.

### Funding
The authors received no funding for this work.

### Competing Interests
The authors declare there are no competing interests.

## Author Contributions

- Siming Zheng conceived and designed the experiments, performed the experiments, analyzed the data, contributed reagents/materials/analysis tools, prepared figures and/or tables, performed the computation work, authored or reviewed drafts of the paper, approved the final draft.
- Rahmita Wirza O.K. Rahmat conceived and designed the experiments, performed the experiments, contributed reagents/materials/analysis tools, authored or reviewed drafts of the paper, approved the final draft, checked the logic of this article.
- Fatimah Khalid conceived and designed the experiments, analyzed the data, contributed reagents/materials/analysis tools, authored or reviewed drafts of the paper, approved the final draft, checked the fluency of this article.
- Nurul Amelina Nasharuddin analyzed the data, contributed reagents/materials/analysis tools, prepared figures and/or tables, performed the computation work, authored or reviewed drafts of the paper, approved the final draft, examined the experimental codes.

## Data Availability

The raw data is available at zheng, siming (2019): 1-step-FRGC2-Original-Dataset-.zip. figshare. Figure. DOI: 10.6084/m9.figshare.9791951.v1.

## Supplemental Information

Supplemental information for this article can be found online at http://dx.doi.org/10.7717/peerj-cs.236#supplemental-information.

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
