# Peer review of "D texture-based face recognition system using fine-tuned deep residual networks"

_PeerJ Computer Science, doi:10.7717/peerj-cs.236_

## Round 0.1 · original submission · Major Revisions

This paper deals with 3D face recognition on what the authors call a large data set. I don't think that around 4000 samples is large enough for the mission. And augmentation is mentioned only in the last section of the paper, surely has not been applied, On the whole, the results look nice.

Reviewer 1 ·

Basic reporting

- English needs to be improved. Some examples where language could be improved are as follows:
- 188: First letters of terms with abbreviation should be capital letters
- 207: Gamma to be written as a symbol
- 268: Singular verb with plural subject
- 58, 243, 273-275, 304: Inappropriate usage of capital letters
- 292-300: Inappropriate usage of past tense
- 416, 487-489: Inappropriate usage of punctation mark
- 229-235: Need to use inline latex scripts

- Figure 2 is not cited in the main text.
- Figure 2 caption refers to the term "Face Recognition Grand Challenge" first time, hence its abbreviation (FRGC) must appear in parenthesis near from it.
- Figures 8, 9 and 10 are parts of a multi-part figure. Please arrange them as they are to be published, label each part with an uppercase letter and submit them in one file.
- When starting a sentence with a citation, use 'Figure 1'. For example in line 166, 258.
- Please include figure numbers in filenames, for example 'Figure 1.png'

- Thanks for providing all figures with a high resolution however your figure titles need to be more descriptive. For example:
- Figure 1 title is "3D face pre-processing.png". Using file name with its extension makes comprehension difficult. In addition preprocessing is only one phase of this figure which causes ambiguity.
- Figure 4 title is: "The HOG features of various person samples". Please consider to use "3D face images" instead of "person samples".
- Figure 5 title is: "The structure of ResNet in different layers". I suggest to use the term "architecture" instead of "structure". Also mentioning different layers is unnecessary.
- Figure 6 title is: "The Fine-Tuned Resnet Neural Network". Inappropriate usage of capital letters.
- Figure 8, 9 and 10's title is: "The accuracy rate of different layer numbers in the fine-tuning ResNet architecture." Please indicate the number of layers explicitly; 50, 101 or 152. Reader can not understand which figure part is for which number of layers.

- Please include table numbers in filenames, for example 'Table 1.png'.
- Table 1 is redundant of the information included in Figure 8, 9 and 10.
- The "Method" column in Table 2 contains reference numbers but these reference numbers can not be interpreted.
- Authors provided the source code however raw dataset, preprocessing code and pre-trained models are missing.

Experimental design

- Research question is defined as achieving high recognition accuracy on a large scale 3D face image dataset in the presence of gradient vanishing problem. However your experiments are on FRGC dataset which contains 4000 images and is not a large scale dataset. On the other hand you use ResNet which has no gradient vanishing problem. I suggest that the knowledge gap being filled in your paper is the usage of HOG features as the input representation for 3d image recognition.

- I could not grasp the parts of method through B3 to B4. For example line 230 says "we evenly divide the vectors into nine intervals" then line 247 talks about small and large intervals. What does "category information" refer to?

- Method's reproducibility requires the followings:
- Gamma value in equation 1 is not defined.
- Which pre-trained weights did you use for fine tuning? From which link can we download the pre-trained weights? Are they pretrained on HOG features of 3D face images?
- Line 400 says that fully connected layer can classify more than 1000 types. But line 355 says that there are 466 people in the dataset.

Validity of the findings

No comment

Reviewer 2 ·

Basic reporting

This paper is clearly not ready for publication.

First, there is the writing. The logical flow of the paper is completely flawed and the paper is not coherent and confusing. I am not even able to understand why this is 3D face recognition if all images and the descriptors are 2D. Some examples: the word "Thus" in the abstract does not follow from the previous sentence, The first paragraph (and the others) of the introduction jumps from one topic to another. The related work section jumps from ideas to almost irrelevant minor details and it also reports recognition results without pointing to the exact benchmark. H is not defined in Eq. 2.

Second, it is not clear what is the claimed novelty of the paper since it mostly relies on existing methods.

Experimental design

The paper needs much work on all parts leading to the experiments that I did not read this section.

Validity of the findings

The paper needs much work on all parts leading to the experiments that I did not read the results section.

Additional comments

Without additional work, the manuscript cannot be published. Focus on making clear and coherent arguments. Clearly state what you do and why it is novel.

---

## Round 0.2 · accepted · Accept

The English has improved a lot. The paper now is ok.

Below are few correctable mistakes which should be addressed while in production.

Good luck

The sentence in lines
67-68 does not parse well especially line 68

180 implemented by uses the open-source, should be
implemented by using the open-source

236 since gamma=1 equation (1) is totally redundant

270-271 bad English

Also there : In some cases, the algorithm eliminates some detailed features, which
are represented by the small amplitude gradient in the intervals. The rest of blocks are merged

Should be:
In some cases, the algorithm eliminates some detailed features, which
are represented by the small amplitude gradient in the intervals. The rest of THE
blocks are merged

337 Such first-layer features are not used to be specific to specific
data sets or specific tasks, but to general ones,
TOTALLY UNCLEAR

340 which can greatly shorten the training time, efficiency

efficiently?

433 Firstly, a convolution layer in fine-tined

Should be
First, ….fine-tuned?

461 equation 8 are all variables described?